# Activation of Porous Pt Electrodes Deposited on YSZ Electrolyte by Nitric Acid Treatment

**DOI:** 10.3390/ma14185463

**Published:** 2021-09-21

**Authors:** Liliya Dunyushkina, Anastasiya Pavlovich, Adelya Khaliullina

**Affiliations:** Laboratory of Electrochemical Material Science, Institute of High Temperature Electrochemistry, 20 Akademicheskaya St., 620137 Ekaterinburg, Russia; nastenka_98@mail.ru (A.P.); adelia01@mail.ru (A.K.)

**Keywords:** electrode activation, electrode polarization, Pt electrode, YSZ electrolyte, nitric acid

## Abstract

The effect of nitric acid treatment on the electrochemical performance of porous Pt electrodes deposited on YSZ (abbreviation from yttria stabilized zirconia) electrolyte was investigated. Two identical symmetrical Pt/YSZ/Pt cells with porous Pt electrodes were fabricated, after which the electrodes of the first cell were kept as sintered, while those of the second cell were impregnated with HNO_3_ solution. The electrochemical behavior of the prepared electrodes was studied using impedance spectroscopy and cyclic voltammetry. Significant reduction of the polarization resistance of the HNO_3_-treated electrodes was revealed. The observed enhancement of the electrochemical performance of porous Pt electrodes was assumed to be caused by adsorption of NO_x_-species on YSZ and Pt surfaces, which promotes oxygen molecules dissociation and transport to the triple phase boundary by the “relay-race” mechanism. The obtained results allow for considering the nitric acid treatment of a porous Pt electrode as an effective way of electrode activation.

## 1. Introduction

Solid oxide fuel cells (SOFCs) are high−efficient electrochemical devices that convert the chemical energy of fuel directly into electrical energy by an environmental-friendly way. In the last decades, significant efforts have been undertaken to commercialize SOFC technologies. A critical challenge in this field is decreasing SOFC operating temperature, which is typically above 800 °C, which would help shorten the startup time, lengthen the lifetime, and reduce the cost. Enhancement of SOFC electrodes activity is important to address this problem as the electrode performance is largely responsible for effective operation of solid oxide cells.

Different ways of electrode activation have been developed. Cathodic polarization was reported to significantly enhance the electrochemical activity of LSCF/SDC and LSM/YSZ composite cathodes [1,2], LSCF cathodes [3], Pt electrodes [4,5,6] (LSCF, SDC, LSM, and YSZ denote La_1−*x*_Sr*_x_*Co_1−*y*_Fe*_y_*O_3−d_, Sm-doped ceria, LaSr_1−*x*_Mn*_x_*O_3−d_, and yttria stabilized zirconia, respectively). However, the polarization-induced activation is generally not durable; the electrode relaxation gradually occurs after interruption of polarization. The impregnation of porous electrodes with solutions of cerium or praseodymium nitrates, which decompose to the related oxides upon heating, was found to be effective for enhancement of the electrode performance [7,8,9,10]. Cerium nitrate solution was successfully used for activation of Ni-cermet SOFC anodes [7,8], whereas the impregnation with a solution of praseodymium nitrate was reported to activate cathodes such as In_2_O_3_ [9] and Pt [10,11]. The observed decrease of the electrode polarization resistance was explained by extension of the oxygen reduction/oxidation reaction area due to infiltration of the oxide particles into the porous electrode matrix [7,8,9,10]. However, the salt solution impregnation results in decreasing the electrode porosity because of formation of oxide particles in the pores, which can lead to slowdown of gas diffusion and passivation of the electrode.

In our recent research devoted to assessing the applicability of the electrochemical amperometric sensor based on YSZ electrolyte with two symmetrical Pt electrodes for measuring small nitric oxide concentrations in nitrogen−oxygen mixtures, it was revealed that the presence of small amounts of NO (below 1 vol.%) in the O_2_-containing gas mixtures resulted in a significant increase of the sensor limiting current [12]. It was assumed that the adsorbed NO species promoted the dissociative adsorption of molecular oxygen and transport of oxygen atoms to the TPB region (TPB is a triple phase boundary between the solid electrolyte, electrode, and gas). High capacity of YSZ to adsorb NO_x_ (NO_x_ stands for NO and NO_2_) species was revealed and explained by participation of oxygen vacancies in the electrolyte which provide NO_x_ adsorption sites [13,14]. It is reasonable to expect that the adsorbed NO_x_ species should affect the oxygen exchange rate at a YSZ/electrode interface.

In the present research, the effect of the NO_x_ species on the polarization resistance of porous Pt electrodes deposited on YSZ electrolyte (ZrO_2_ doped by 9 mol.% Y_2_O_3_) was investigated. The porous electrodes were impregnated with nitric acid. It is known that nitric acid when it is exposed to light or high temperature decomposes according to the following reaction:4HNO_3_ → 4NO_2_↑ + 2H_2_O + O_2_↑(1)

Upon heating above 320 °C nitrogen dioxide decomposes as [15]:2NO_2_ → 2NO + O_2_.(2)

So, the nitric acid impregnated into the porous electrode acts as a source of nitrogen oxides on heating, that should enhance the oxygen exchange at the YSZ/Pt interface. If that is confirmed, then the impregnation of porous Pt electrodes with nitric acid can be used as a simple and cost-effective way of the electrode activation.

For comparison, two symmetrical Pt/YSZ/Pt cells with the as-prepared electrodes (cell 1) and the impregnated with HNO_3_ electrodes (cell 2) were fabricated. The morphology of the porous Pt electrodes was studied by scanning electron microscopy (SEM). Raman spectroscopy was applied to characterize the YSZ surface after the HNO_3_ treatment. Thermogravimetry (TG) and differential scanning calorimetry (DSC) was used to characterize the mass losses of YSZ sample wetted with HNO_3_ solution upon controlled heat treatment and to analyze the evolved gases. Electrochemical performance of the differently treated electrodes was investigated using impedance spectroscopy and cycling voltammetry. The origin of the activation effect of the HNO_3_ treatment of a porous Pt electrode deposited on YSZ electrolyte was discussed.

## 2. Materials and Methods

Two symmetrical electrolyte-supported Pt/YSZ/Pt cells with porous Pt electrodes were fabricated as follows. Square plates of YSZ with dimensions 0.8 cm × 0.8 cm × 0.05 cm were made by hot casting and sintered at 1650 °C. The electrode slurry was prepared by mixing dispersed Pt powder with ethanol. The slurry was painted symmetrically onto the opposite sides of YSZ plates and sintered at 900 °C for 2 h in air. The electrode area was about 0.4 cm^2^. The overall Pt loading for each electrode in the studied cells was approximately 7.5 mg cm^−2^. The electrodes of the first cell were kept as-prepared (cell 1), whereas those of the second cell were impregnated with ~5 μL g^−1^ of HNO_3_, 70% (cell 2).

To study the effect of the amount of nitric acid impregnated into the porous Pt electrode on its activity, three similar cells were prepared as described above, but the amount of HNO_3_ solution varied as ~5, ~10, and ~15 μL g^−1^.

The microstructure of the electrodes was studied by means of scanning electron microscopy (SEM) using MIRA 3 LMU (Tescan, Brno, Czech Republic). For microstructural investigations, a cross section of a fractured YSZ/Pt sample was examined.

To characterize the effect of HNO_3_ treatment on the state of YSZ surface, Raman spectroscopy was applied. For these measurements, two pieces of a fractured YSZ plate, one of which was kept intact and the other was immersed in a nitric acid bath, lifted, and heated to 800 °C, were prepared. The spectra were measured in a reflection mode using a Renishaw U 1000 microscope-spectrometer (Stonehouse, UK) equipped by a 532 nm Nd:YAG laser at room temperature. The laser power at the sample was about 50 mW.

Thermogravimetry (TG) and differential scanning calorimetry (DSC) were used to characterize the mass losses of YSZ powder wetted with HNO_3_ solution upon heating from 35 to 1100 °C. For these measurements, a piece of YSZ ceramics was grinded to a powder in a leucosapphire mortar, wetted with HNO_3_ solution and dried at room temperature in air. Then, 18.5 mg of the obtained powder were analyzed using STA 449 F1 Jupiter (NETZSCH, Selb, Germany) and mass spectrometer QMS 403 C Aeolos (NETZSCH, Germany) in air flow of 20 mL min^−1^, at the heating rate of 10 °C min^−1^. 

The electrochemical performance of the cells was investigated by means of impedance spectroscopy and cyclic voltammetry (CV) using a Parstat 2273 potentiostat (Advanced Measurement Technology Inc., Oak Ridge, TN, USA). The cells were installed on a test stand having rooms for two cells to insure identical testing conditions. Platinum meshes with a size of 5 × 5 mm^2^ were used as current collectors. For connection of the cells with the external circuit, platinum wires were attached to the current collectors. The cells were heated at a ramp rate of 50 °C/h to 800 °C in air in order to decompose nitric acid, and then cooled to 600 °C. As far as the electrode performance depends on the fabrication process and testing history, cell 1 with the as-prepared electrodes was thermally treated in the same way as cell 2. The impedance measurements were carried out in the frequency range 0.1 Hz–1 MHz with an amplitude of 30 mV and resolution of 30 points per decade in the temperature range 600–800 °C with a step of 50 °C in air. CV testing of the cells was performed at 600 °C. The applied DC voltage was swept forth and back between zero and ±2.0 V with a scan rate of 0.5 mV/s. At several values of DC voltage (±0.5, ±1.0, ±1.5, and ±2.0 V), the impedance of the cells was measured under the corresponding applied DC bias.

## 3. Results

### 3.1. Sample Characterization

Figure 1 shows the SEM micrograph of a cross section of YSZ/Pt sample. As can be seen, the electrode thickness is about 20 μm; the structure is porous and consists of fine and coarse platelet-like particles ranging in size from the submicron level to several micrometers.

The Raman spectra of the pristine and HNO_3_ treated YSZ samples presented in Figure 2 are similar in the region of Raman shift below 1700 cm^−1^. For the acid-treated sample, additional features can be seen in the range of 1700–3500 cm^−1^, namely, small intensity bands at about 1710, 1750, 1875, 1970 cm^−1^, and 3040 cm^−1^; besides, the intensity of the band at 2285 cm^−1^ increased strongly. The Raman bands observed at wavenumbers below 700 cm^−1^ are typical for YSZ structure [16]. According to [17], the Raman mode of NO and NO_2_ are observed at 1875 cm^−1^ and 1316 cm^−1^, respectively. So, the small intensity band at 1875 cm^−1^ on the Raman spectrum of the HNO_3_ treated sample can be related to nitric oxide. The similarity of the spectra for the pristine and acid-treated samples at the wavenumbers of around 1316 cm^−1^ indicates that nitrogen dioxide decomposes according to the reaction (1) upon the thermal treatment at 800 °C.

High capacity of YSZ to adsorb NO_x_ species (NO_x_ stands for NO and NO_2_) was reported in [13,14]. It was assumed that the interaction of NO_x_ on YSZ surface happens with the participation of oxygen vacancies in the electrolyte, which provide NO_x_ adsorption sites. The data on temperature−programmed desorption (TPD) and diffuse reflectance infrared Fourier transform spectroscopy (DRIFTS) reported in [14] indicate that NO*_x_* interaction with YSZ is strong, forming different types of nitrates: bridging monodentate, chelating, and bridging bidentate. It is reasonable to assume that the additional Raman peaks observed for the HNO_3_ treated YSZ sample can be caused by these species; however, further study is needed for exact assignment of all Raman bands.

TG and DSC curves obtained for the nitric acid treated YSZ powder are shown in Figure 3a. The total mass loss in the temperature range from 35 to 1100 °C achieves 1.38%. The changes of ionic currents presented in Figure 3b indicate that the sample loses H_2_O (*m*/*z* = 18), CO (*m*/*z* = 28), CO_2_ (*m*/*z* = 44), NO (*m*/*z* = 30), and NO_2_ (*m*/*z* = 46) in the investigated temperature range. The main weight decrease (1.16%) occurs upon heating to ~600 °C. The weight loss is accompanied by endo−effect at 577.8 °C indicating nitrogen monoxide removal from the sample that is confirmed by the mass-spectrometric analysis of 30 *m*/*z* ion formation. At higher temperatures, the sample continues to lose mass due to the slow evolution of water, carbon oxides, and nitrogen monoxide. Thus, according to the obtained TG-DSC data, in the temperature range of 600–800 °C, in which the electrochemical properties of the Pt/YSZ/Pt cells were examined, the adsorbed NO species persist on YSZ surface, and hence can affect the cell electrochemical performance.

### 3.2. Electrochemical Testing

The electrochemical performance of the Pt/YSZ/Pt cells was investigated by means of impedance spectroscopy and cyclic voltammetry. First, the cells were heated to 800 °C in order to burn out nitric acid, and then cooled to 600 °C for the measurements. For illustration, the Nyquist plots of the cells 1 and 2 at 600 °C are given in Figure 4. The intercept of the impedance spectra at high frequencies corresponds to the electrolyte resistance, R_YSZ_. The electrode polarization resistance was determined as the difference of the intercepts of impedance spectra with real axis at high frequency and low frequency ranges. As can be seen, the cell 2 with the electrodes treated with HNO_3_ demonstrates much smaller polarization resistance than the cell 1.

The results of CV testing of the cells 1 and 2 was performed at 600 °C are shown in Figure 5. During the run, the electrochemical impedance responses under the DC bias values of ±0.5, ±1.0, ±1.5, and ±2.0 V were measured. The discontinuities observed on the voltage−current density curves are caused by these measurements. As can be seen in the insert in Figure 5, the electrode polarization resistance decreases as the DC voltage increases for the both cells; but the cell 2 with the HNO_3_-treated electrodes exhibits the smaller polarization resistance in the whole range of the applied voltages. Figure 6 shows the electrochemical impedance responses of the cells 1 and 2 measured under zero bias before and right after the CV testing. As can be seen, the electrode polarization resistance of both cells significantly decreased after the testing. 

The impedance spectra of the cells in the temperature range of 600–800 °C were measured after the CV testing. For illustration, the Nyquist plots at 650 and 800 °C are presented in Figure 7. The Arrhenius plots of the electrode polarization resistance are shown in Figure 8. As can be seen, the polarization resistance depends strongly on the electrode treatment: cell 2 with the electrodes treated with HNO_3_ demonstrates much smaller polarization resistance than cell 1, and the discrepancy increases as temperature decreases. The activation energy of the electrode polarization resistance of cell 1 is about 70 ± 5 kJ/mol, which is close to the reported value for the Pt/YSZ system (80 ± 5 kJ/mol) [18]. The HNO_3_-treated electrodes show the smaller activation energy of 49 ± 4 kJ/mol. The electrochemical testing of the Pt/YSZ/Pt cells with the as-prepared and nitric acid treated electrodes, which lasted for about three weeks, showed that the benefits of the acid treatment continued after the exposure to high temperature (up to 800 °C) and applying significant DC voltage (up to 2 V).

To examine the effect of the amount of nitric acid impregnated into the porous Pt electrode on its activity, three similar cells were prepared as described above, but the electrodes were treated with ~5, ~10, and ~15 μL g^−1^ of HNO_3_ solution. The cells were studied by impedance spectroscopy. The Nyquist plots for these cells measured at 600 °C are presented in Figure 9. As can be seen the electrode resistance decreases significantly upon adding 5 μL g^−1^ HNO_3_ and tends to saturation upon increasing the nitric acid amount. The Nyquist plots for these cells at 600 °C are presented in Figure 9. As can be seen the electrode resistance decreases significantly upon adding 5 μL g^−1^ HNO_3_ and tends to saturation upon increasing the nitric acid amount. So, the optimal amount of nitric acid lies between 5 and 10 μL g^−1^.

## 4. Mechanism of the Electrode Activation

In spite of the fact that the zirconia−based electrolyte/Pt electrode system is considered as a model one, the mechanism of the oxygen exchange reaction in this system is still not fully understood. Typically, the oxygen exchange at the Pt/YSZ interface is assumed to proceed as a sequence of several steps which includes the dissociative adsorption (or desorption) of oxygen molecules at the Pt and YSZ surfaces, the surface diffusion of oxygen atoms to the TPB region, the oxygen reduction (or oxidation) reaction, and the incorporation (or release) of oxygen ions into YSZ lattice. The dissociative adsorption and surface diffusion of oxygen are usually considered to be the rate−determining step of the oxygen exchange kinetics at the Pt/YSZ interface at high temperatures [18,19,20,21]. 

One can assume that the nitric acid treatment of porous Pt electrodes enhances the rate of these steps. In accordance with the reactions (1,2), the impregnation of the porous Pt electrode with nitric acid followed by annealing at 800 °C results in the formation of nitric oxide, which can adsorb on the surfaces of the electrode and the electrolyte. As it was shown in [13,14], YSZ possesses high capacity to adsorb nitrogen oxides due to the presence of oxygen vacancies which act as adsorption sites. Formation of the bridged nitrate species on YSZ surface due to the strong interaction of NO with YSZ should facilitate the dissociative adsorption and surface diffusion of oxygen. NO molecules adsorbed on Pt surface being a free radical promote the dissociative adsorption of oxygen and associate with up to two oxygen ions forming nitrite or nitrate ions. The energy of the O–NO_2_ bond (~150 kJ/mol [22]) is much less than the dissociation energy of the oxygen molecule (498 kJ/mol); so, the preferred pathway of oxygen ions over Pt surface to the TPB region is the transport between the adsorbed NO_3_^−^ and NO_2_^−^ ions by the “relay-race” mechanism as it was suggested in [12].

Thus, application of the simple and cost-effective technological operation based on impregnation of the porous Pt electrode with nitric acid allows to effectively reduce the electrode polarization resistance.

## 5. Conclusions

The influence of nitric acid treatment of porous Pt electrodes deposited on YSZ electrolyte on the electrochemical performance was investigated. TG−DSC data indicate that in the temperature range of 600–800 °C, in which the electrochemical properties of the Pt/YSZ/Pt cells were examined, the adsorbed NO species persist on YSZ surface. Raman spectroscopy study showed the appearance of new Raman peaks for the HNO_3_ treated YSZ sample, which can be caused by formation of different types of nitrates. Electrochemical testing of the Pt/YSZ/Pt cells using impedance spectroscopy and cyclic voltammetry showed that nitric acid impregnation of porous Pt electrodes resulted in the significant reduction of the polarization resistance. The observed enhancement of the electrochemical performance of porous Pt electrodes was assumed to be caused by adsorption of NO_x_-species on YSZ and Pt surfaces, which promotes oxygen molecules dissociation and transport of oxygen atoms to the TPB by the “relay-race” mechanism. Despite the necessity of further study of the mechanism of the electrode activation, the obtained results allow for considering the treatment of porous Pt electrodes with nitric acid as an effective way of the electrode performance enhancement.

## Figures and Tables

**Figure 1 materials-14-05463-f001:**
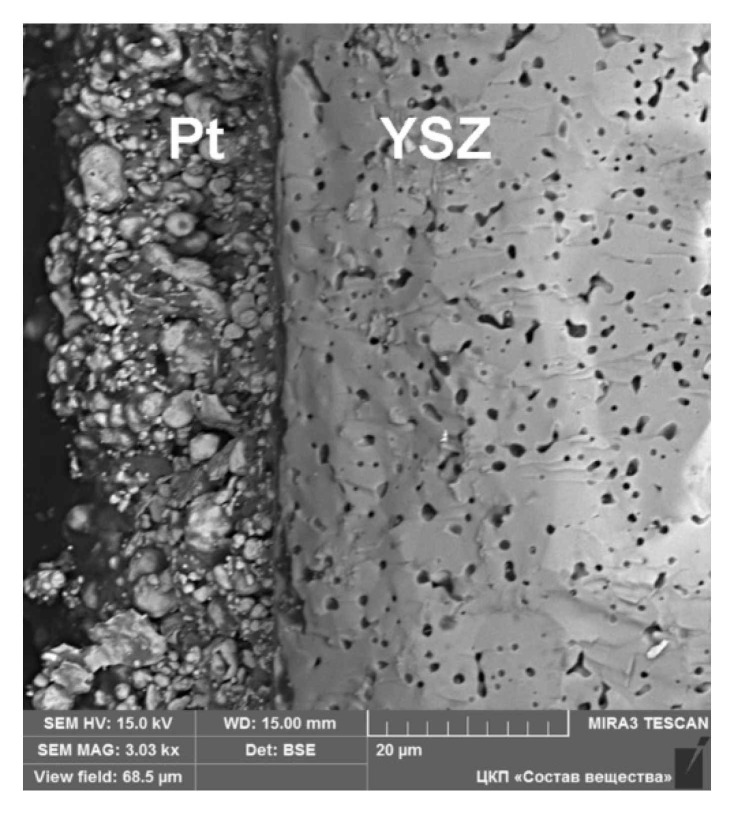
SEM image of a cross section of YSZ/Pt sample.

**Figure 2 materials-14-05463-f002:**
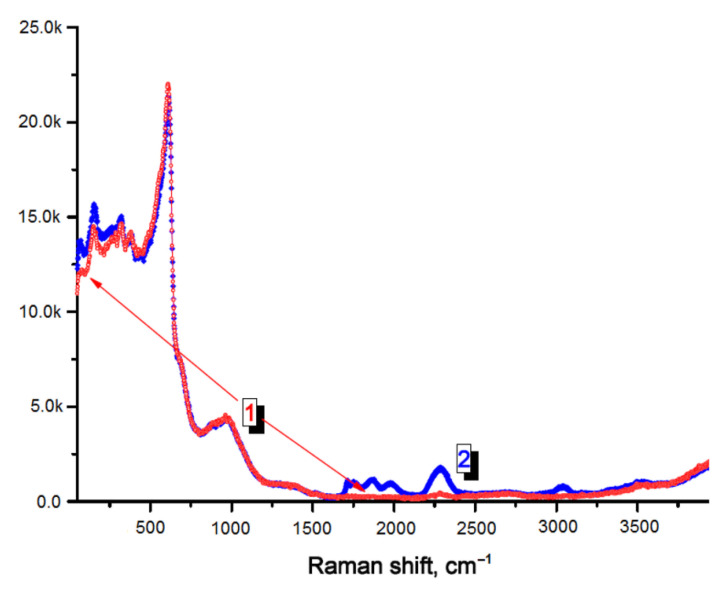
Raman spectra of the pristine (1, red) and HNO_3_ treated (2, blue) YSZ ceramic plates.

**Figure 3 materials-14-05463-f003:**
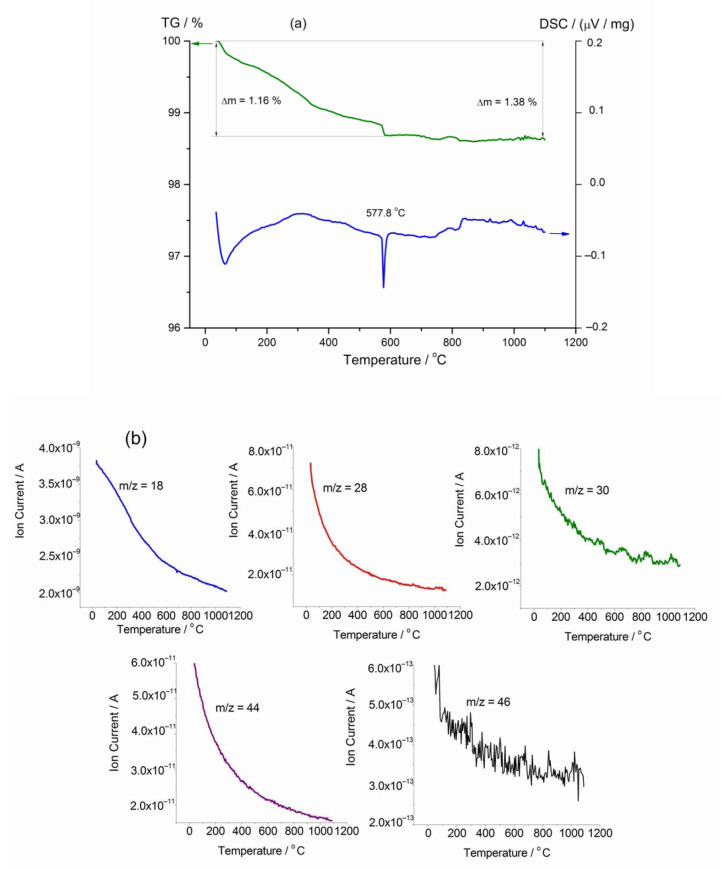
TG-DSC (**a**) and ion current data (**b**) for YSZ powder wetted with HNO_3_ solution. Mass-to-charge ratios, *m*/*z*, correspond to H_2_O (18), CO (28), NO (30), CO_2_ (44), and NO_2_ (46).

**Figure 4 materials-14-05463-f004:**
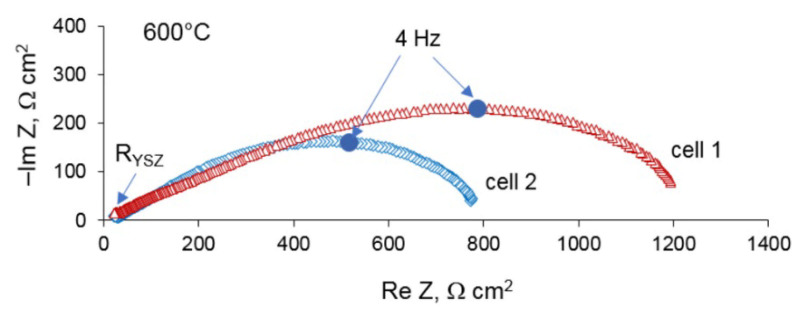
Impedance spectra of the cells 1 and 2 at 600 °C.

**Figure 5 materials-14-05463-f005:**
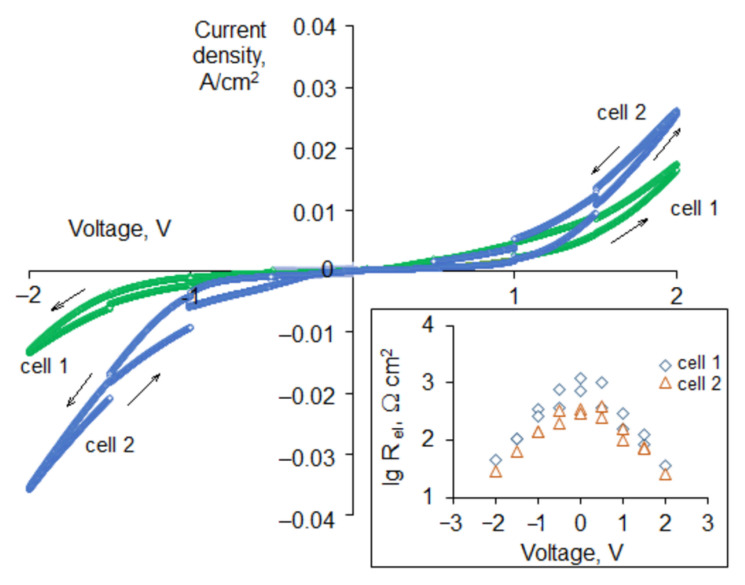
Cyclic voltammetry curves of cells 1 and 2 at 600 °C. In the insert: electrode polarization resistance as a function of applied DC voltage.

**Figure 6 materials-14-05463-f006:**
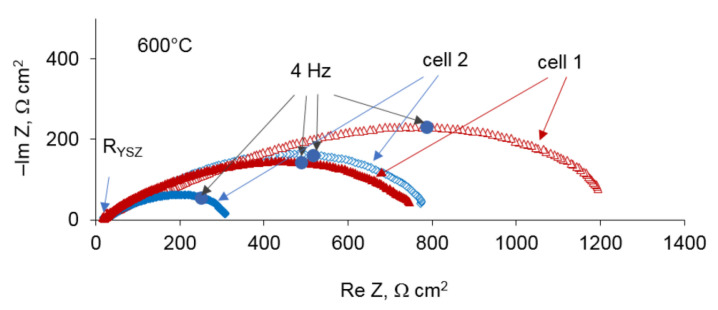
Impedance spectra of cells 1 and 2 before (open symbols) and after (closed symbols) CV testing at 600 °C.

**Figure 7 materials-14-05463-f007:**
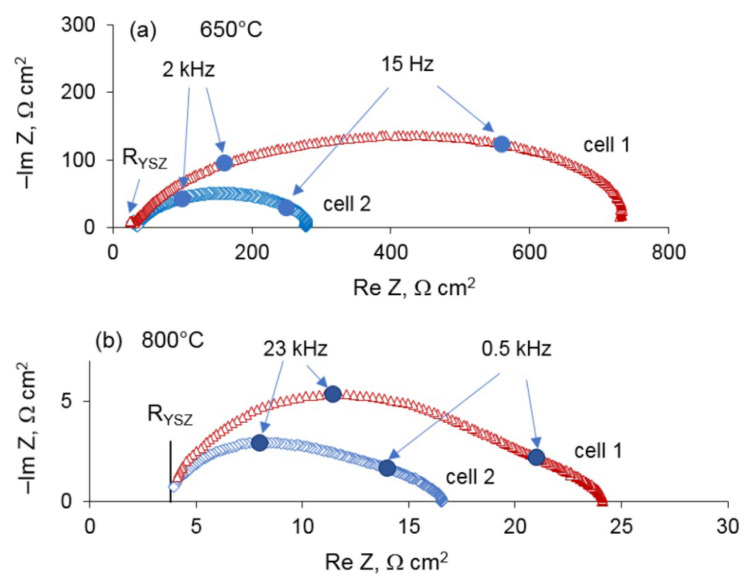
Impedance spectra of cells 1 and 2 at (**a**) 650 and (**b**) 800 °C.

**Figure 8 materials-14-05463-f008:**
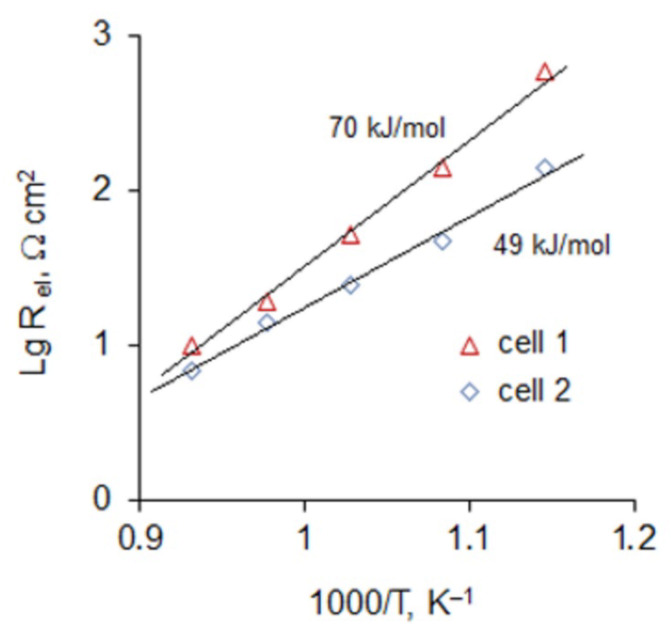
Arrhenius plots of the electrode resistance of cells 1 and 2.

**Figure 9 materials-14-05463-f009:**
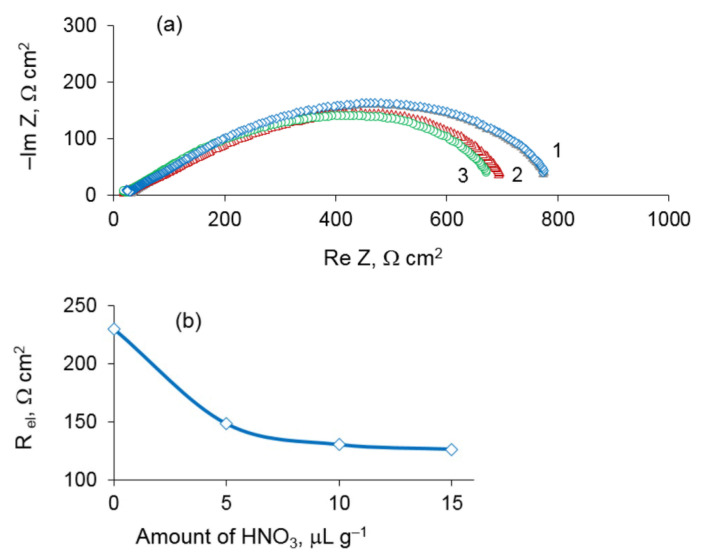
Impedance spectra of the cells Pt/YSZ/Pt with electrodes impregnated by 5 (1), 10 (2) and 15 mL g^−1^ HNO_3_ at 600 °C (**a**) and electrode polarization resistance of Pt/YSZ/Pt cell vs. amount of HNO_3_ (**b**).

## Data Availability

Not applicable.

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
