# Peer review of "Activation of Porous Pt Electrodes Deposited on YSZ Electrolyte by Nitric Acid Treatment"

_materials, 2021, doi:10.3390/ma14185463_

Round 1
Reviewer 1 Report
A good manuscript describing a useful method and material. It contains, however, several technical and formal flaws so that minor revision is mandatory. If these are eliminated, the manuscript will be much more attractive. In more detail:
- The Introduction part contains trivial statements. Note that readers are experts who know this. They will be bored when reading this. The authors do not adequately explain why a reader of the article should use the new method rather than an existing method that possibly works quite well and is simpler.
- The abbreviation should be given when you use the term first.
- Does the author use any fitting for impedance spectroscopy? If yes, provide the circuit model.
- Inset of Figure 9 author used the alphabetic term a, b. but in figure caption use numerical value 1, and 2. Author should check and correct the mistake.
Author Response
First of all, the authors would like to thank the Reviewer for the constructive and insightful comments to our work. We believe that the Reviewer evaluation of the manuscript provides us with an opportunity to make our results clearer and more useful for readers. In the revised manuscript, the corrections are given in green to help the Editor and the Reviewers to find the revised parts easily. We outline below our response to the Reviewer comments.
Reviewer 1
A good manuscript describing a useful method and material. It contains, however, several technical and formal flaws so that minor revision is mandatory. If these are eliminated, the manuscript will be much more attractive. In more detail:
Comment: The Introduction part contains trivial statements. Note that readers are experts who know this. They will be bored when reading this. The authors do not adequately explain why a reader of the article should use the new method rather than an existing method that possibly works quite well and is simpler.
Reply: Thanks for your kind suggestion. We suppose that it is reasonable to briefly review the known methods of electrode activation, their advantages and drawbacks in Introduction, as far as this research is devoted to the development of a new activation technique. We have enhanced the advantages of the suggested technique in Introduction.
Comment: The abbreviation should be given when you use the term first.
Reply: Correction was made.
Comment: Does the author use any fitting for impedance spectroscopy? If yes, provide the circuit model.
Reply: In the frame of this research, we examined the changes of the total polarization resistance of the Pt/YSZ system, which was determined as the difference of the intercepts of impedance spectra with real axis at high frequency and low frequency ranges, upon HNO3-treatment. Detailed analysis of the electrode polarization behavior is supposed to be the subject of future research.
Comment: Inset of Figure 9 author used the alphabetic term a, b. but in figure caption use numerical value 1, and 2. Author should check and correct the mistake.
Reply: The mistake was corrected.
Reviewer 2 Report
The paper cen be published after minor revision reflecting comments inserted as yellow notes into attached pdf of submitted manuscript

Author Response
First of all, the authors would like to thank the Reviewer for the constructive and insightful comments to our work. We believe that the Reviewer evaluation of the manuscript provides us with an opportunity to make our results clearer and more useful for readers. In the revised manuscript, the corrections are given in green to help the Editor and the Reviewers to find the revised parts easily. We outline below our response to the Reviewers’ comments.
Reviewer 2
Comment: The paper can be published after minor revision reflecting comments inserted as yellow notes into attached pdf of submitted manuscript.
Reply: Corrections were made.